# Performance evaluation of the *diaxxoPCR* system for rapid and user-friendly stool-based diagnosis of trichuriasis, ascariasis and strongyloidiasis in Mozambique

**Augusto Messa Jr.** [1,2,3]* , **Michel Bengtson** [4] , **Pedro Fleitas** [2] , **Luca Montemartini** [5,6] , **Valdemiro Novela** [1] , **Áuria de Jesus** [1] , **Alejandro Krolewiecki** [7,8] , **Tobias Schindler** [5,6] , **Jose Muñoz** [2,3,9] , **Inácio Mandomando** [1,2,10,11] , **Lisette van Lieshout** [4]

**1** Centro de Investigaçao em Saúde da Manhiça (CISM), Maputo, Mozambique, **2** ISGlobal, Barcelona, Spain, **3** Facultat de Medicina i Ciències de la Salut, Universitat de Barcelona (UB), Barcelona, Spain, **4** Leiden University Center for Infectious Diseases (LUCID), Parasitology Research Group, Leiden University Medical Center (LUMC), Leiden, The Netherlands, **5** Institute for Chemical and Bioengineering, ETH Zurich, Zuerich, Switzerland, **6** Diaxxo AG, Zuerich, Switzerland, **7** Universidad Nacional de Salta, Instituto de Investigaciones de Enfermedades Tropicales/CONICET, Oran, Salta, Argentina, **8** Mundo Sano, Buenos Aires, Argentina, **9** International Health Department, Hospital Clínic de Barcelona, Barcelona, Spain, **10** Instituto Nacional de Saúde (INS), Maputo, Mozambique, **11** Global Health and Tropical Medicine, GHTM, Associate Laboratory in Translation and Innovation Towards Global Health, LA-REAL, Instituto de Higiene e Medicina Tropical, IHMT, Universidade NOVA de Lisboa, UNL, Lisboa, Portugal

☙ These authors contributed equally to this work.
* augusto.junior@manhica.net

## Abstract

### Background

Nucleic acid amplification tests have shown promising results for the diagnosis of soil-transmitted helminths (STH). The implementation of real-time PCR (qPCR) in low-resource settings is, however, still hampered by multiple logistical challenges. In this study, we assessed the diagnostic performance of a cartridge-based real-time PCR system (*diaxxoPCR*) for the detection of DNA of *Trichuris trichiura*, *Ascaris lumbricoides,* and *Strongyloides stercoralis* in clinical samples, using qPCR as the reference test.

### Methodology

Initially, a technical validation of the *diaxxoPCR* system in a singleplex pod (cartridge) design was performed using 37 predefined DNA samples (Study A), followed by a diagnostic comparison between the *diaxxoPCR* system (singleplex) and qPCR on DNA samples extracted from 325 stools collected in a clinical trial in a rural area of Mozambique (Study B). Finally, one negative and one positive DNA sample were used to demonstrate the technical performance of a multiplex pod design as a potential use-case for the *diaxxoPCR* system (Study C).

**Data availability statement:** All relevant data are in the manuscript and its Supporting Information files.

**Funding:** This project was funded at CISM and the LUMC by the EDCTP2 program supported by the European Union (grant number RIA2017NCT-1845 -STOP; www.stoptheworm.org) Horizon 2020 European Union Funding for Research and Innovation (JM, LvL, AK and IM). The funders had no role in the design of the study, data collection, analysis, decision to publish, or preparation of the manuscript.

**Competing interests:** We have read the journal's policy and the authors of this manuscript have the following competing interests: LM and TS are shareholders of the ETH Zurich spin-off company Diaxxo AG. The authors declare that the study was conducted in the absence of commercial or financial relationships that could interfere with the results or interpretation.

## Principal findings

Study A demonstrated that the *diaxxoPCR* system performed reliably for each of the three STH targets, with minimal intra- and inter-assay variation and sufficient output reproducibility. Study B, performed in Mozambique, showed a positive qPCR result in 57.5% (187), 15.4% (50), and 0.3% (1) of the 325 DNA trial samples for *T. trichiura*, *A. lumbricoides,* and *S. stercoralis*, respectively. The *diaxxoPCR* system demonstrated sensitivities and specificities above 97% and 94% for each target, resulting in nearly perfect to perfect qualitative agreements with the reference test. Quantitatively, significant and positive associations were seen between the Ct-values (qPCR) and Cq-values (*diaxxoPCR*). In Study C, the *diaxxoPCR* system correctly detected all 3 targets in the multiplex pod design.

## Conclusion

With refinements regarding faecal DNA extraction procedures, the *diaxxoPCR* system has potential to provide accurate and easy-to-use real-time molecular diagnostics of STH in low resource laboratories.

### Author summary

Intestinal parasitic worms remain an important cause of disease in low- and middle-income countries, particularly in tropical regions with poor access to adequate water, sanitation, and hygiene. Microscopy of stools is the cornerstone to assess the necessity and the success of control programs but is limited in use mainly due to the need for expert microscopists and its shortcomings in diagnostic accuracy. Molecular diagnostics, including real-time PCR (qPCR), offer a highly sensitive and specific alternative, but accessibility in poor-resource settings remains a challenge. In this study we evaluated a portable and affordable cartridge-based real-time PCR system (*diaxxoPCR*) for the diagnosis of 3 intestinal parasitic worms and compared its performance to a well-established qPCR protocol. Overall, the *diaxxoPCR* system produced qualitative and quantitative results that were analogous to those obtained with qPCR, which shows that it has the potential to improve the accessibility of molecular diagnostic procedures in poor-resource settings, where these parasites are a public health problem. The *diaxxoPCR* system can be adapted for the purpose of STH species-specific population-based surveys or the management of individual patients.

## 1. Introduction

Soil-transmitted helminths (STH) are intestinal parasitic worms including *Trichuris trichiura* (whipworm), *Ascaris lumbricoides* (roundworm), hookworms (*Necator americanus* and *Ancylostoma duodenale*) and *Strongyloides stercoralis* (threadworm)

[1]. Although they are more frequent in areas with poor access to adequate water, sanitation, and hygiene (WASH) in low- and lower-middle-income countries, they also occur in high-income countries, particularly in vulnerable populations, and are associated with malnutrition, impaired growth, and cognitive development in children [1,2]. The 2021 Global Burden of Disease study estimated around 643 million (586 – 715 million) STH infections, with 1.38 million (0.91 – 2.02 million) DALYs lost [3]. This excludes strongyloidiasis, a disease caused by *S. stercoralis*, which is still neglected despite being clinically relevant due to the risk of hyperinfection syndrome in immunosuppressed/immunocompromised individuals and estimated to affect 386 million (324 – 449 million) people [1,4,5]. The World Health Organization (WHO) recommends the administration of benzimidazole drugs (albendazole or mebendazole) in mass-drug administration (MDA) campaigns targeting pre-school (pre-SAC) and school-aged children (SAC), women of childbearing age and adults in certain high-risk occupations, in what is called Preventive Chemotherapy (PC) [1,6]. However, the efficacy of this regimen is limited in the case of *T. trichiura* and *S. stercoralis* infections [4].

It has long been recognised that diagnosis plays not only a crucial role in the clinical management of individual patients, but also in the control of STH, since highly accurate diagnostic tools are essential for making evidence-based decision on intervention programs and (dis)continuation of treatment schedules [7]. A clear illustration of this need is in the WHO 2030 roadmap for STH, where both target #1 (achieve and maintain elimination of STH morbidity in pre-SAC and SAC) and target #2 (reduce the number of tablets needed in PC), require surveys to study the epidemiology of STH following at least 5 years of PC to establish whether they have been achieved or not [8,9]. The WHO recommends the Kato-Katz (KK) thick stool smear for the detection and quantification of the intensity of infections however, despite being adequate for mapping STH infections and initiating PC programs, it may be insufficient for stopping PC-decision and post-PC surveil-lance [8,9]. Nucleic acid amplification tests (NAAT), particularly real-time polymerase chain reaction (qPCR), have been shown to be a valuable alternative [8,10–12]. However, despite accurate diagnostic performances, the general adoption of qPCR in STH endemic regions faces many logistical challenges, including the technical complexity of the procedure, the need for advanced laboratory infrastructure, the high costs of reagents and the accessibility of materials [12–14]. Some of these challenges can in principle be addressed by financial support, e.g., upgrading laboratory facilities, provid-ing training schemes for technicians, and subscribing to external quality assessment programs [13,15]. But particularly in sub-Saharan countries the difficulty of purchasing laboratory reagents, certainly when a cold chain is required, remains an important obstacle for the full implementation of qPCR for the diagnosis of STH.

An easy-to-use, portable qPCR platform combined with a cartridge system would be an attractive alternative, addressing several of the challenges highlighted above. The *diaxxoPCR* system is designed to meet the requirements for making qPCR more accessible for the remote areas of endemic countries where a more user-friendly approach is mostly needed, by performing the identification, quantification, and genotyping of pathogens, requiring minimal hands-on operations and small reagent and sample volume [16]. Additionally, it eliminates cold-chain requirements during the reagents' shipping, storage, or usage since the reactions are run in aluminium-based cartridges, which come preloaded with all reagents in dried form. The current iteration of the cartridges/pods is equipped with 20 wells, each accommodating a single qPCR assay to measure one sample. Finally, it can be used in mobile testing applica-tions, powered by a car battery, and results can be accessed directly on the device's screen or via a standard web browser on a mobile phone or laptop, requiring only that both devices are connected to the same local network, even without internet access. No software installation is required, as all data analysis and processing are handled directly on the *diaxxoPCR* device itself.

The *diaxxoPCR* system has been evaluated for the detection of SARS-CoV-2 variants of concern [17] and for the quantitative detection of *Plasmodium* spp. [18], but not for STH. Herein, we report on the performance of the *diaxxoPCR* system, as an easy-to-perform, rapid, and highly sensitive NAAT-based diagnostic tool for STH, focusing on *T. trichiura*, *A. lumbricoides*, and *S. stercoralis*. We compare it to a currently available qPCR using DNA extracted from stools collected in children and young adults enrolled in a clinical trial for the evaluation of safety and efficacy of a fixed-dose co-formulation

of Albendazole and Ivermectin (ALIVE) in the Manhiça district, Southern Mozambique (NCT05124691) between October 2022 and March 2023. Additionally, a potential use-case in a diagnostic laboratory scenario is explored.

## 2. Materials and methods

### 2.1. Ethics statement

The ALIVE trial protocol was approved by the Institutional Bioethics Committee for Health at the Centro de Investigação em Saúde de Manhiça (CIBS-CISM), the National Bioethics Committee for Health (CNBS – IRB00002657), and the Autoridade Nacional Reguladora de Medicamentos (ANARME) in Mozambique. Before any study activities, written informed consent was obtained from the participants or their parents or guardians (for those under 18 years). Additionally, a signed informed assent was obtained if children were 12 years or older. The ALIVE trial was registered at ClinicalTrials.gov (NCT05124691).

### 2.2. Study design

The performance evaluation study was conducted in two laboratories, at the Centro de Investigação em Saúde da Manhiça (CISM), Maputo, Mozambique and the Leiden University Medical Centre (LUMC), Leiden, The Netherlands. The study was carried out in three parts: A, B, and C (Fig 1).

Study A was a technical laboratory-based validation study with four components, conducted to assess (i) intra-user variation, (ii) day-to-day variation, (iii) diagnostic accuracy and (iv) inter-user variation between institutes of the *diaxxoPCR*

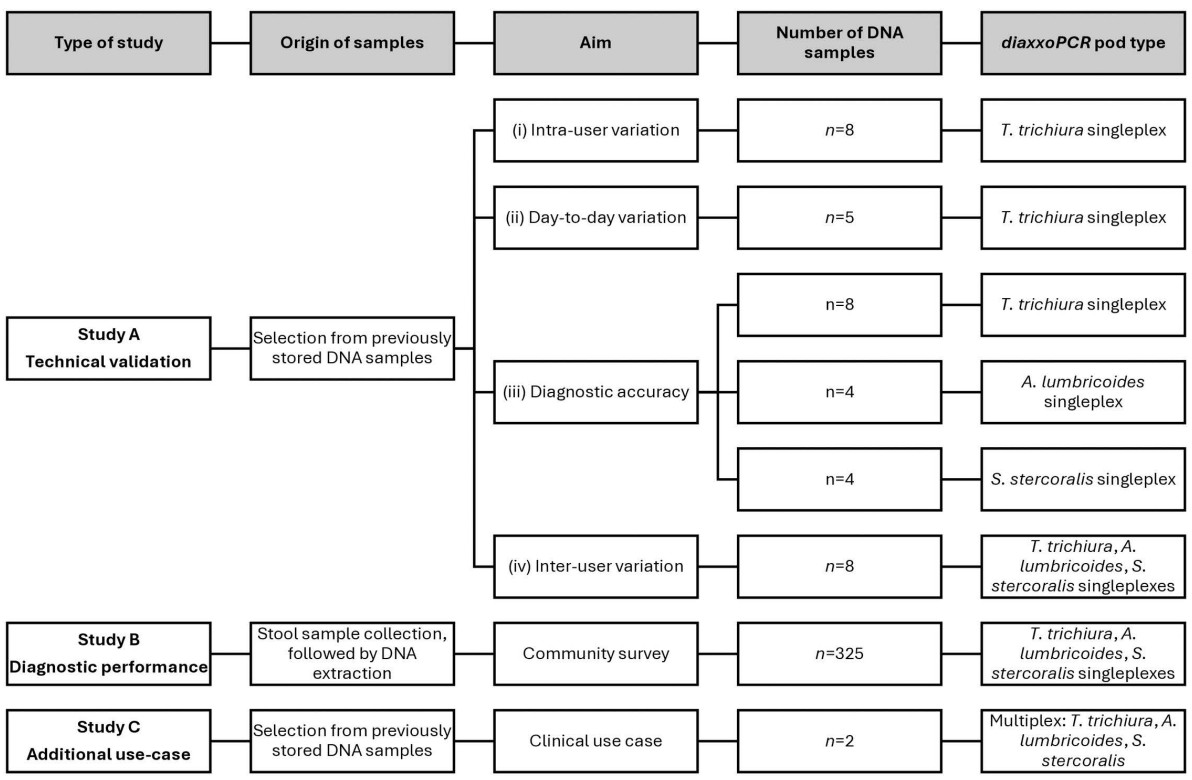

**Fig 1. Flowchart depicting the study design and sample selection for the performance evaluation of the *diaxxoPCR* system.** *Samples for study B were collected as part of a clinical trial.

system. For components (i) and (ii), the focus was on the performance of the *T. trichiura diaxxoPCR* pods, while for components (iii) and (iv), *A. lumbricoides* and *S. stercoralis* pods were also included. All pods used in Study A had a single target design (singleplex). Samples were selected from previously characterised human stool DNA aliquots that were either distributed within the EDCTP-funded STOP project for qPCR quality control purposes or were leftovers from participation in the international Helminth External Molecular Quality Assessment Scheme (HEMQAS) [15]. These DNA samples were selected based on known microscopy egg counts in stool (only known for *T. trichiura*) and/or qPCR results, i.e., which was not restricted to *T. trichiura*, *A. lumbricoides* and *S. stercoralis*, but also included *N. americanus*, *A. duodenale* and *Schistosoma spp.* This ensured that a mix of positive and negative samples were included in the technical validation (unblinded). For all these 5 targets, the qPCR assays were performed at the LUMC, with the outcome listed in Sheets A, B, C and D in S1 Table. Due to the availability of these stored DNA aliquots, components (i) to (iii) of the *diaxxo*PCR were performed at the LUMC, while component (iv) was performed at both institutes.

Based on the available volume of different samples for Study A and the number of wells on the pods, the number of DNA samples tested for the different components had to be limited. To assess the intra-user variation (i), a single user tested 8 DNA samples, all from *T. trichiura* microscopy and qPCR-positive stool samples, in 3 *diaxxoPCR* runs on the same day. To assess day-to-day variation (ii), 5 DNA samples, isolated from 4 *T. trichiura* microscopy and qPCR positive stools and 1 negative, were tested by the same single *diaxxoPCR* user across 4 different days. To assess the diagnostic accuracy (iii), 11 DNA samples, qPCR positive for *T. trichiura* ($n = 8$), *A. lumbricoides* ($n = 4$), and *S. stercoralis* ($n = 4$), were tested on the *diaxxoPCR* system for these targets using singleplex pods for each species. Finally, to assess inter-user variation (iv), 8 DNA samples, qPCR positive for different helminth targets, were tested at each institute on the *diaxxoPCR* system for *T. trichiura A. lumbricoides* and *S. stercoralis* (all singleplex).

To provide a more realistic field-based diagnostic performance assessment, Study B was conducted at CISM. This study was an independent (blinded) diagnostic performance evaluation using 325 stored DNA samples extracted from stool samples collected in Mozambique during the ALIVE Clinical Trial, described in detail elsewhere [19,20]. Briefly, ALIVE was a multicentric trial performed from January 2022 to March 2023 to evaluate the safety and efficacy of a fixed-dose combination (FDC) of Albendazole and Ivermectin in children and young adults (5–18 years). In Mozambique, participants for the trial were enrolled between October 2022 and March 2023 in three schools in a semi-rural district located approximately 80 km north of the country's capital city Maputo in the southern region of the country, after informed consent was obtained.

Study C was carried out at the LUMC as a proof-of-concept evaluation to preliminarily demonstrate the technical performance of a multiplex pod design as a potential use case. For this purpose, two DNA samples were selected, analogous to Study A. One sample was known to be positive by qPCR for *T. trichiura*, *A. lumbricoides* and *S. stercoralis*, with a Ct-value of 27.4, 21.3 and 27.1 for these 3 targets, respectively. The other sample was known to be qPCR negative for the same three targets. Both samples were tested in triplicate on the *diaxxoPCR* multiplex pod.

## 2.3. Sample collection and preparation

The description of the sample collection methods for Study B has been published elsewhere [19,20]. In brief, after stool sample retrieval from participants, they were transported to CISM's laboratory in a cooler box within 4 hours, where they were assessed for the presence of STH eggs and larvae by microscopic examination, using the Kato-Katz and the modified Baermann [21] techniques, respectively. Participants positive for STH who met ALIVE eligibility criteria were then enrolled, randomized, and treated. Twenty-one±7 days after treatment, a follow-up stool sample was collected following the same method for assessment of treatment efficacy according to the study protocol. Stool samples from participants who were positive at the screening visit and all samples from the follow-up visit were aliquoted in non-denatured absolute ethanol (Sigma-Aldrich, Darmstadt, Germany, cat: 1085430250) and then stored at -20ºC until DNA extraction was performed. For Study B, samples from both the baseline and follow-up visits were included in the diagnostic performance assessment.

## 2.4. Diagnostic methods

**2.4.1. Nucleic acid extraction and real-time PCR.** Genomic DNA extraction of the ALIVE samples (Study B) was performed in batches of 24 samples, including one tube with phosphate-buffered saline – PBS (Gibco by Life Technologies, Grand Island, NY, USA, cat. 20012–027) as a no template control per batch. The extractions were performed using a QIAamp DNA Mini Kit (Qiagen, Hilden, Germany, cat. 51306) following the manufacturer's protocol with minor modifications. Briefly, 250 µL of the ethanol-preserved faeces suspension was transferred to a 2 mL PowerBead tube containing 1.4 mm ceramic beads (Qiagen, cat. 13113050), centrifuged at 14000 × g for 1 minute, then the ethanol supernatant was discarded, and the pellet washed with 1000 µL of PBS, then centrifuged and the supernatant discarded. Thereafter, 200 µL of 2% polyvinylpolypyrrolidone – PVPP (Supelco by Sigma-Aldrich, Saint Louis, MO, USA, cat. 77627) was added to the tubes for inhibitor adsorption. This was followed by bead-beating for 10 min at 50 Hz in a Mini-beadbeater-16 (BioSpec Products, Bartlesville, OK, USA, cat 607EUR). Tubes were frozen at -80°C for 30 min, then brought back to room temperature, vortexed, and incubated for 10 minutes at 100°C. Tubes were spun quickly, followed by DNA extraction following the QIAamp DNA Mini Kit manufacturer's instructions, except for using 400 µL AL buffer spiked with Phocid Herpesvirus-1 – PhHV (European Virus Archive – EVAg, cat. 011V-00884) as an internal control. Eluted DNA (200 µL) was stored at 2–8°C until qPCR and *diaxxoPCR* experiments, which were all conducted within 6 months of DNA extraction (August 2023 – February 2024). The stored DNA samples used for studies A and C were extracted by the same QIAamp DNA Mini Kit procedure, including the same PhHV internal control and a bead-beating step.

The ALIVE DNA samples (Study B) were tested at CISM in duplicate in 2 qPCR multiplex panels, ANAS (*A. duodenale*, *N. americanus*, *A. lumbricoides,* and *S. stercoralis*) and ST (*Schistosoma* spp. and *T. trichiura*). Primers and probes targeting PhHV as an internal control were included in both panels. Oligonucleotide sequences and concentrations for the detection of STH have been published elsewhere [22]. Experiments were performed on an Applied Biosystems 7500 Real-Time PCR System (Applied Biosystems, Foster City, CA, USA, cat: 4351105). A sample was considered positive for a specific target in the qPCR if an amplification curve was observed with a Cycle threshold (Ct) -value ≤35 in the two replicates and negative when both Ct-values were >35. Samples were repeated if there was a discrepancy between the replicates or if they showed a difference in Ct-value >3.3 units. The stored DNA samples used for Study A and Study C have been tested in the same 2 qPCR multiplex panels at the LUMC, however run on a CFX real-time detection system (Bio-Rad laboratories). According to the established procedure at the LUMC, the samples were tested a single time and were considered positive when showing an amplification with a Ct-value <40 [22].

**2.4.2. Description of the *diaxxoPCR* system.** The experiments on the *diaxxoPCR* system were performed using the cartridges (pods) supplied by Diaxxo AG (Zuerich, Switzerland). The 20 well cartridges contain all reagents necessary for running the qPCR in preloaded and dried form. The STH assays are TaqMan probe-based qPCR assays that use a 6-Carboxyfluorescein (6-FAM) labelled probe to detect a specific amplification product during PCR. The same oligonucleotide sequences and concentrations for the detection of STH for qPCR were used, except for the modification of the labelled probes [22]. For Studies A and B, three sets of singleplex cartridges were designed to target *T. trichiura*, *A. lumbricoides*, and *S. stercoralis*, whereas for Study C multiplex cartridges were used, targeting all three species in a single pod.

Experiments were conducted following the steps described in the *diaxxoPCR* instructions for use (S1 File). In each run, 18 wells were loaded with 4.5 µL of DNA eluate and the 2 other wells were loaded with a no template control (nuclease-free water) and a positive control (DNA from adult parasites) for each respective species. The following cycling parameters were executed on the *diaxxoPCR* instrument: incubation for 300s at 53˚C, followed by initial polymerase activation for 60 seconds at 94˚C and then 45 cycles of 10 seconds at 94˚C for denaturation and 20 seconds at 60˚C for extension. Raw data was analysed by the *diaxxoPCR* system integrated software, providing a Cq-value as the Cycle threshold output, which were automatically assigned to the samples. Specimens with amplification with a Cq-value <40 were considered positive.

## 2.5. Data analysis

Individual sample results from experiments were entered in a Microsoft Excel Spreadsheet, double-checked and then results were summarized using descriptive statistics, summary tables (e.g., 2x2) and graphs. Diagnostic performance metrics, including the sensitivity, specificity, positive predictive value (PPV), negative predictive value (NPV) and the corresponding 95% confidence intervals of the *diaxxoPCR* system, were calculated for *T. trichiura*, *A. lumbricoides*, and *S. stercoralis* detection, using qPCR as the reference standard. Qualitative agreement between qPCR and *diaxxoPCR* was assessed using the adjusted Cohen's kappa, considering true positives and true negatives, as well as false positives and false negatives [23]. The strength and direction of the association between Ct-value (qPCR) and Cq-value (*diaxxoPCR*) between the two methods was evaluated using Spearman's correlation coefficient ($\rho$), excluding the negative data points. All statistical analysis and data visualization was performed using the R statistical language (version 4.3.3) [24].

## 3. Results

### 3.1. Study A: Technical validation of the *diaxxoPCR* system

To test for intra-user variation, 8 samples positive for *T. trichiura*, with known egg counts ranging in EPG (eggs per gram of faeces) from 456 to 10,068, were tested on the *diaxxoPCR* system. The *diaxxoPCR* system correctly detected all these 8 component (i) samples as *T. trichiura* positive, although some variation in Cq-values was observed between the 3 runs. Table 1 summarizes the outcomes of the technical validation of component (i), while the individual sample data can be found in Sheet A in S1 Table. In component (ii), the *T. trichiura diaxxoPCR* system correctly detected all 4 positive samples and showed no signal for the negative sample across all four days of assessment (Table 1; for individual sample data, see Sheet B in S1 Table). In component (iii), the *T. trichiura diaxxoPCR* system correctly detected 6 positive and 2 negative samples, and the *A. lumbricoides* and *S. stercoralis diaxxoPCR* pods correctly detected each of the 4 positive samples (for individual sample data, see Sheet C in S1 Table). In component (iv), *T. trichiura diaxxoPCR* showed full agreement in 7 of the 8 samples. One sample showed a positive *diaxxoPCR* result both at the LUMC and CISM, with exactly the same Cq-value (35.2), while being negative by qPCR. The *A. lumbricoides diaxxoPCR* showed agreement in 6 of the 8 samples. One sample showed a positive *diaxxoPCR* result at the LUMC (Cq-value 28.0), and another sample showed a positive *diaxxoPCR* result at CISM, while both these samples were negative by qPCR. The *S. stercoralis diaxxoPCR* showed agreement in 7 of the 8 samples. Similar to the *T. trichiura diaxxoPCR*, a positive *diaxxoPCR* result was seen both at the LUMC (Cq-value 30.6) and CISM (Cq-value 23.6), while the qPCR was negative. None of the qPCR-positive samples were missed by any of the *diaxxoPCR* pods. Individual sample data for component (iv) are shown in Sheet D in S1 Table. Based on these results, we proceeded with Study B at CISM.

**Table 1.** Outcomes of technical validation of the diaxxoPCR in comparison to qPCR of T. trichiura in Study A components (i) and (ii) using stored DNA samples.

| Diagnostic test | Study A component i (*n*=8) | | | Study A component ii (*n*=5) | | | | | |
| | Microscopy | qPCR | *diaxxoPCR* | Microscopy | qPCR | *diaxxoPCR* | | | |
| | | | | | | Day 1 | Day 7 | Day 14 | Day 21 |
| Positives (*n*) | 8 | 8 | 8 | 4 | 4 | 4 | 4 | 4 | 4 |
| Negatives (*n*) | 0 | 0 | 0 | 1 | 1 | 1 | 1 | 1 | 1 |
| Output | EPG | Ct-value | Cq-value | EPG | Ct-value | Cq value | Cq value | Cq value | Cq value |
| Range* | 456-10,068 | 23.1-30.0 | 16.7-31.5 | 444-10,068 | 23.1-30.0 | 17.2-24.8 | 21.5-32.3 | 17.7-24.9 | 16.7-24.7 |
| Median* | 1,314 | 27.4 | 23.6 | 1,572 | 27.4 | 20.5 | 21.5 | 20.2 | 18.8 |

EPG – Eggs per gram of stool; * – only for positive samples.

### 3.2. Study B: Diagnostic performance of the *diaxxoPCR* system on samples from the ALIVE clinical trial

Tested by qPCR, a positive signal was seen in 187 (57.5%), 50 (15.4%) and 1 (0.3%) of the 325 DNA trial samples for *T. trichiura*, *A. lumbricoides* and *S. stercoralis*, respectively. Qualitatively, nearly perfect agreements were observed between the *diaxxoPCR* system and qPCR for *T. trichiura* and *A. lumbricoides*, and a perfect agreement for *S. stercoralis* (Table 2). The sensitivity of the *diaxxoPCR* system was above 97%, and the specificity was above 94% for each of the three targeted species. The *diaxxoPCR* system missed 4 of the 187 samples that tested positive by qPCR for *T. trichiura* but detected all qPCR-positive samples for both *A. lumbricoides* (n = 50) and *S. stercoralis* (n = 1). Among the 4 *T. trichiura* samples with discrepant results, 3 were from the post-treatment visit, 2 were also microscopy negative and with high Ct-values by qPCR; one had a low intensity of infection (96 EPG) and the other had moderate intensity (5412 EPG). The *diaxxoPCR* system detected 8 and 12 samples that tested negative by qPCR as positives for *T. trichiura* and *A. lumbricoides*. Among the 8 false positives for *T. trichiura*, one was negative by microscopy (post-treatment visit), and the other seven were positive by microscopy, with 12–24 EPG. Among the 12 *A. lumbricoides* false negatives, 6 were microscopy negative, and the remaining 6 were positive with light (4 samples, range: 144–2256 EPG) and moderate infections (2 samples with ≥24000 EPG). No samples were misidentified for *S. stercoralis*. Individual sample data for Study B are shown in Sheet A in S2 Table.

Significantly strong and positive associations were observed between the Ct-values for qPCR and Cq-values from the *diaxxoPCR* system for both *T. trichiura* (Fig 2a) and *A. lumbricoides* (Fig 2b). The association between the two NAAT procedures was not assessed for *S. stercoralis* because there was just one sample positive for this species, which was detected by both methods.

### 3.3. Study C: Potential clinical use case at LUMC

In Study C, the *diaxxoPCR* system correctly detected all 3 targets in all the triplicates of the positive sample, with Cq-values comparable to the qPCR Ct-value. The average Cq-value and standard deviation were 27.3 ± 0.12 for *T. trichiura*, 22.2 ± 0.97 for *A. lumbricoides* and 28.4 ± 1.24 for *S. stercoralis*. The *diaxxoPCR* also classified the negative sample correctly in all 3 replicates. Individual sample data for Study C are shown in Sheet B in S2 Table.

## 4. Discussion

In this study, we evaluated for the first time the performance of the *diaxxoPCR* system, an easy-to-use, portable qPCR platform with no cold-chain requirements, for the diagnosis of the STH species *T. trichiura*, *A. lumbricoides* and *S. stercoralis* in DNA samples extracted from human stool samples. Performed in a laboratory situated in an STH endemic region, the *diaxxoPCR* system demonstrated high sensitivity, specificity, and good qualitative and quantitative agreement compared to qPCR as the reference standard for the three STH targets under evaluation. qPCR was used as the reference test because currently, STH diagnosis lacks a true "gold standard" [25].

The study was designed as a proof-of-principle to determine whether cartridge-based NAAT diagnosis could be performed on stool samples collected within the context of the ALIVE clinical trial, which took place at CISM in Mozambique,

**Table 2. Diagnostic performance of the *diaxxoPCR* system compared against qPCR for the detection of STH species in 325 stool samples collected during the ALIVE clinical trial in Manhiça district, Southern Mozambique (Study B).**

| Species | Sens (CI) | Spec (CI) | PPV (CI) | NPV (CI) | Cohen's kappa | Cohen's kappa interpretation |
|---|---|---|---|---|---|---|
| *T. trichiura* (N = 325) | 97.9 (95.8-99.9) | 94.2 (90.3-98.1) | 95.8 (93-98.7) | 97 (94.1-99.9) | 0.92 (0.88-0.97) | Near perfect |
| *A. lumbricoides* (N = 325) | 100 (100-100) | 95.6 (93.2-98.1) | 80.7 (70.8-90.5) | 100 (100-100) | 0.87 (0.8-0.94) | Near perfect |
| *S. stercoralis* (N = 325) | 100 (100-100) | 100 (100-100) | 100 (100-100) | 100 (100-100) | 1 (1-1) | Perfect |

Sens – Sensitivity; Spec – Specificity; PPV – Predictive Positive Value; NPV – Negative Predictive Value; CI – 95% Confidence Interval.

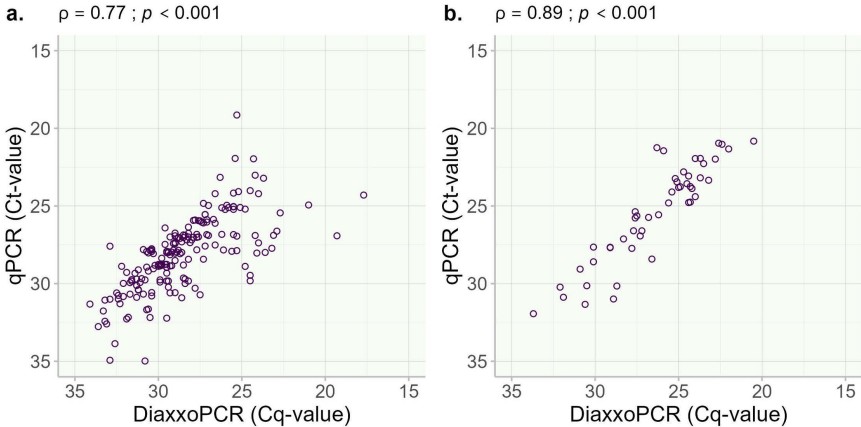

**Fig 2. Scatter plots illustrating the association between Ct-value by qPCR and Cq-value by *diaxxoPCR* for a) T. trichiura and b) A. *lumbricoides*.** ρ (rho) – Spearman's correlation coefficient.

as well as in Kenya and Ethiopia [19,20]. In this trial, stool samples were collected to study the therapeutic efficacy of a coformulation of albendazole and ivermectin. The beneficial effect of the treatment was mainly expected for the parasites *T. trichiura* and *S. stercoralis*, hence the inclusion of these two targets in the *diaxxoPCR*. Within the boundaries of the budget and available labour at the laboratory, one more DNA target could be included in the study. *A. lumbricoides* was selected, as microscopy data suggested this parasite to be sufficiently present in the sample collection of Study B. Hookworm was not included as a *diaxxoPCR* target for several reasons, it includes multiple species (*N. americanus* and *Ancylostoma spp.*) and therefore it would take at least double the amount of cartridges, microscopy data showed it to be hardly present in the collection of Study B, and, finally, we noticed some technical challenges in the preparation of the cartridges targeting the hookworm species, which could not be fully solved within the time and budget boundaries of the project.

For the technical validation (Study A), we focused comparatively more on the detection of *T. trichiura* than on the two other species. This decision was made because we had access to sufficiently well-characterised DNA samples with *T. trichiura* microscopy data to further explore the reproducibility of the performance of the cartridges of the *diaxxoPCR* system. In addition, *T. trichiura* is acknowledged to be the most challenging of the STH species when implementing an NAAT. This is mainly due to the robustness of *T. trichiura* eggs, which hampers optimal DNA extraction [22]. Here, we used the same DNA samples, all obtained via a well-established column-based procedure, including a rigorous bead-beating step and the inclusion of an internal control [22]. We observed high levels of concordance between the qPCR and the *diaxxoPCR* system when testing for intra-user variation, day-to-day variation and diagnostic accuracy (components i to iii), while some qualitative discrepancies were seen when comparing the performance of the *diaxxoPCR* system between LUMC (Netherlands) and CISM (Mozambique). In total, the *diaxxoPCR* showed a positive signal 6 times, which was not observed at the qPCR. This happened 3 times at the LUMC and 3 times at CISM, of which 4 times, the *diaxxoPCR* at the LUMC and at CISM had similar results which could not be confirmed by the qPCR. In one sample, the *T. trichiura* Cq-values of the *diaxxoPCR* at the two laboratories were just above 35, indicating a low concentration of this target, which could explain why the qPCR was negative. In the other sample, the *S. stercoralis diaxxoPCR* Cq-values were substantially below 35, while the qPCR showed no positive signal for this target. As the same DNA sample has been used for each NAAT, contamination during the extraction procedure seems unlikely. qPCR showed this sample to be highly positive for *T. trichiura* and *A. lumbricoides*, which suggests some non-specific crossover output of the *diaxxoPCR* system. However, this is not supported by the findings of study B, where the *S. stercoralis* target was found to be 100% specific. Quantitatively, some variation was noticed in the Cq-value when testing was repeated (components i and ii), but this variation

seemed to be random. Variation in the quantitative output of different NAAT systems has been described before, even when using the same DNA sample [15,26].

In Study B, a strong correlation was seen between *diaxxoPCR* (Cq-value) and qPCR (Ct-value). The number of qualitative discrepancies between *diaxxoPCR* and qPCR were low and generally seen in samples of low intensity of infection, where reduced reproducibility of both microscopy and NAAT is generally acknowledged [22]. Similarly to the qualitative outcome, the strength of association of the quantitative output is within the expected range and comparable to what has been described before when comparing Ct-values/Cq-values with stool microscopy [10,13,22]. Again, it confirms what has been elegantly shown via the implementation of an external quality assessment scheme for NAAT-based diagnosis of helminths in stool samples. The variation in quantitative outcome of different NAAT procedures is known to be substantial (up to 10 cycles), not only depending on the used DNA extraction procedure, but also when comparing different NAAT systems on the same DNA samples [15,26]. Recently, it has been suggested that the variation in the quantitative output of NAAT for the detection of helminth targets in stool could be mitigated by the introduction of an international standard of genome equivalents per mL (GE/mL) per DNA target, which could be introduced in each assay in a standard-dilution series [13]. For the *diaxxoPCR* system, the cartridge system with 20 wells per cartridge would be less suitable for the inclusion of such a standard curve per run. And, in general, this does not solve the problem that in clinical samples, helminth eggs will represent a wide variation in maturation and, therefore, will contain substantial differences in the number of target copies per egg [13].

A major advantage of our study is the fact that we had access to a large collection of well-characterized DNA samples collected as part of the ALIVE clinical trial and could perform the analysis within the laboratory setting of CISM (Mozambique). This means we could evaluate the performance of the *diaxxoPCR* system in a setting representative of where observational studies and clinical trials are typically conducted, and more importantly, where many of the practical challenges to implementing qPCR in routine diagnostics are encountered. On the other hand, the prevalence of *S. stercoralis* in the study population was as low as observed in a previous study in the same location, but relatively low compared to other areas in the Manhiça district [27], either based on microscopy (Baermann procedure), qPCR or the *diaxxoPCR* system. We would therefore like to repeat the detection of *S. stercoralis* by the *diaxxoPCR* in a setting with higher prevalence and we would also like to include additional targets such as hookworm and *Schistosoma.* Furthermore, the DNA extraction procedure used included the addition of an internal control in the form of a non-human virus (PhHV), which was targeted in the qPCR for potential detection of inhibition [22]. If the qPCR is to be replaced by the *diaxxoPCR* system, ideally controlling of potential inhibition should be included as well, which will increase the number of required cartridges and consequently will lead to higher costs.

Despite the advantages of NAAT for the diagnostics of STH, its performance is highly dependent on the efficiency of the DNA extraction procedure [28,29]. These procedures are still expensive and labour intense, so to make cartridge-based DNA detection accessible in poor-resource settings, a more simplified DNA extraction procedure will be essential. Fortunately, more researchers are aware, and simplified procedures are currently being explored [30]. So far, most of the published research aims towards field-friendly extraction of *Schistosoma* DNA from urine and gynaecological samples, but simplified DNA extraction from stool for the detection of helminths is also under development [22,31].

In this study, we evaluated the CE-certified *diaxxoPCR* system (version M1), which was originally developed and validated for the diagnosis of SARS-CoV-2. Although the system is commercially available, its application outside its original intended use is still under development. Previous work has extended its use to the detection of SARS-CoV-2 variants of concern [17] and the quantitative detection of *Plasmodium* spp. [18]. Here, we assessed the performance of the *diaxxoPCR* system and customised cartridges for STH diagnostics, where it demonstrated good performance. From an operational perspective, there are still issues to be addressed with the supply chain, technical support and device maintenance. Currently, the team can provide online support to deal with any software issues, however, any maintenance that involves hardware requires the device to be shipped back to the factory and to the user. This is expected to improve with the newest iterations and when the market for the system is better established with product representatives serving larger regions,

for example, in Africa and Latin America. In addition, an integrated system (*diaxxoCare*) that performs both extraction and qPCR without the need for human intervention and reducing the time for sample processing has been developed and can be evaluated for diagnostics of STH, including the evaluation of DNA extraction. In general, there is limited capability for instrument technical support by individual companies. Although well-established equipment suppliers offer instrument support to ensure sustainable testing, this service increases the overall costs as well.

The *diaxxoPCR* system is a promising diagnostic assay that is simple to use and produces results in ~34 minutes for STH (users' observation). The results are automatically processed and saved on the device, with a Cq-value and a visual colorimetric readout displayed on the screen. In addition, the results saved in the device can be assessed and exported using a web-browser-based interface through a local wi-fi connection, even without internet or remotely through an internet connection. These attributes combined with the high sensitivity and specificity demonstrated in this study meet (at least partially) the key specifications of diagnostic technologies that Stuyver and Levecke highlighted as requirements to achieve Target #1 from the WHO 2030 targets for STH [8]: (i) provide information on STH-attributable morbidity, (ii and iii) generate quantitative readout for the STH species separately (multiplexing), (iv) have a clinical sensitivity of at least 95% for moderate and high-intensity infections but similar to single KK for low-intensity infections, and (v) clinical specificity equal or superior of a single KK in individuals with moderate and high-intensity infections. Though not addressed in this study, requirements to achieve Target #2 are theoretically achievable since the only requirement is for the technology to be integrated with the program decision process, i.e., results generated with the *diaxxoPCR* system should be fed directly to the STH control program. The *diaxxoPCR* system has the advantage (at least over current qPCR protocols) of not requiring a cold chain for reagent transportation or storage and requiring minimal hands-on time for sample preparation, in addition to not requiring procurement of individual consumables. Although we did not conduct a direct comparison of the costs to perform both assays, in a previous evaluation, it was estimated that testing with the *diaxxoPCR* system would have a lower cost compared to standard qPCR per target per sample [18]. This advantage was attributed to the lower reagent volumes required for the assays [18]. An evaluation of the costs to conduct the two assays (each with independent DNA extractions) is warranted, particularly in poor-resource settings.

Finally, in Study C, we evaluated the cartridges in a multiplex format, demonstrating the flexibility in pod designs. Although only tested as a proof-of-principle, the format looks promising and could potentially be used as a multiplex cartridge per individual patient sample, thus making qPCR accessible for individual case management in a clinical setting.

## 5. Conclusions

We have evaluated the performance of an easy-to-use and affordable cartridge-based NAAT for STH diagnostics, which showed comparable diagnostic performance as a reference qPCR assay. In many countries where STH are endemic and molecular diagnostics are needed the most, NAAT are either unavailable or only offered at centralized reference laboratories. In this study, we showed that the *diaxxoPCR* system has the potential to bridge this gap and enable increased accessibility in low-resource laboratories to high-quality, sensitive and user-friendly molecular STH diagnostics. Following further refinements, and simplification of DNA extraction procedures, the *diaxxoPCR* system could be used as a diagnostic tool for species-specific population-based surveillance in control programs or to assist clinical trials, and could also be adapted to assist diagnostic management of individual patients.

## Supporting information

**S1 Checklist. STARD-2015 Checklist.** Template source: https://www.equator-network.org/reporting-guidelines/stard/. (DOCX)

**S1 File. *DiaxxoPCR* system instructions for use.** (PDF)

**S1 Table. Individual results for all samples in the technical validation study (Study A).**
(XLSX)

**S2 Table. Individual results qPCR, *diaxxoPCR* and microscopy in the diagnostic performance evaluation (Study B and Study C).**
(XLSX)

## Acknowledgments

The authors would like to acknowledge the ALIVE trial participants (children and parents) from Mozambique for their participation in the study, and the Manhiça district health and education authorities. We would like to acknowledge the ALIVE trial study team in Mozambique (supervisors, doctors, nurses, laboratory technicians, data officers, field officers, and drivers) for their tireless efforts.

CISM receives core funding from the Agencia Española de Cooperacion Internacional para el Desarollo (AECID). ISGlobal authors acknowledge support from the Spanish Ministry of Science and Innovation and State Research Agency through the "Centro de Excelencia Severo Ochoa 2019–2023" Program (CEX2018-000806-S), the CIBER-Consorcio Centro de Investigación Biomédica en Red-(CB 2021), and the Generalitat de Catalunya through the CERCA Program. The use of the PCR internal control PhHV was supported by the European Virus Archive goes Global (EVAg) project that has received funding from the European Union's Horizon 2020 research and innovation program under grant agreements No 653316 and No 871029.

The ALIVE trial was conducted by the STOP project consortium, which includes: Jose Muñoz, Lisette van Lieshout, Alejandro J. Krolewiecki, Charles Mwandawiro, Rachel Pullan, Inácio Mandomando, Maria Martinez Valladares, Wendemagegn Enbiale Yeshaneh, Jaime Algorta, Rafael Guille, Nana Aba Williams, Rafael Balana Fouce, Marc Fernandez, Adelaida Sarukhan, Helena Marti, Berta Grau Pujol, Javier Gandasegui, Valdemiro Novela, Stella Kepha, Martin Rono, Ellie Baptista, Graham Medley, Catherine Pitt, Augusto Messa Jr, Maria Cambra Pelleja, Woyneshet Gelaye, Arthur Nderitu, Paul Gichuki, Carlos Mwandembo, Leah Musyoka, Michel Bengtson, Almudena Legarda, Nuria Cortes, Hilary Smith, Rachel Pullan, Maria Cambra, Miguel Martínez and Steven Doyle.

## Author contributions

**Conceptualization:** Augusto Messa Jr., Michel Bengtson, Lisette van Lieshout.

**Data curation:** Augusto Messa Jr., Michel Bengtson.

**Formal analysis:** Augusto Messa Jr., Michel Bengtson.

**Funding acquisition:** Alejandro Krolewiecki, Jose Muñoz.

**Investigation:** Augusto Messa Jr., Michel Bengtson, Valdemiro Novela, Áuria de Jesus.

**Methodology:** Augusto Messa Jr., Michel Bengtson.

**Project administration:** Valdemiro Novela, Áuria de Jesus, Inácio Mandomando, Lisette van Lieshout.

**Resources:** Tobias Schindler, Lisette van Lieshout.

**Supervision:** Inácio Mandomando, Lisette van Lieshout.

**Validation:** Pedro Fleitas, Lisette van Lieshout.

**Visualization:** Augusto Messa Jr., Michel Bengtson, Lisette van Lieshout.

**Writing – original draft:** Augusto Messa Jr., Michel Bengtson, Lisette van Lieshout.

**Writing – review & editing:** Augusto Messa Jr., Michel Bengtson, Pedro Fleitas, Luca Montemartini, Alejandro Krolewiecki, Tobias Schindler, Jose Muñoz, Inácio Mandomando, Lisette van Lieshout.

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
