## [Decision Letter · Decision Letter 0]

10 Oct 2025

Performance evaluation of the diaxxoPCR system for rapid and user-friendly stool-based diagnosis of trichuriasis, ascariasis and strongyloidiasis in Mozambique

Dear Dr. MESSA,

Thank you for submitting your manuscript to PLOS Neglected Tropical Diseases. After careful consideration, we feel that it has merit but does not fully meet PLOS Neglected Tropical Diseases's publication criteria as it currently stands. Therefore, we invite you to submit a revised version of the manuscript that addresses the points raised during the review process.

Please submit your revised manuscript within 30 days, by November 7th. If you will need more time than this to complete your revisions, please reply to this message or contact the journal office at plosntds@plos.org. Please include the following items when submitting your revised manuscript:

* A rebuttal letter that responds to each point raised by the editor and reviewer(s). You should upload this letter as a separate file labeled 'Response to Reviewers '. This file does not need to include responses to any formatting updates and technical items listed in the 'Journal Requirements' section below.

* A marked-up copy of your manuscript that highlights changes made to the original version. You should upload this as a separate file labeled 'Revised Manuscript with Track Changes '.

* An unmarked version of your revised paper without tracked changes. You should upload this as a separate file labeled 'Manuscript '.

We look forward to receiving your revised manuscript.

Kind regards,

Angela Monica Ionica, Ph.D.

Academic Editor

Jong-Yil Chai

Section Editor

Shaden Kamhawi

co-Editor-in-Chief

Paul Brindley

co-Editor-in-Chief

**Journal Requirements:**

1) Thank you for including an Ethics Statement for your study. Please include:

i) A statement that formal consent was obtained (must state whether verbal/written) OR the reason consent was not obtained (e.g. anonymity). NOTE: If child participants, the statement must declare that formal consent was obtained from the parent/guardian.].

2) Some material included in your submission may be copyrighted. According to PLOSu2019s copyright policy, authors who use figures or other material (e.g., graphics, clipart, maps) from another author or copyright holder must demonstrate or obtain permission to publish this material under the Creative Commons Attribution 4.0 International (CC BY 4.0) License used by PLOS journals. Please closely review the details of PLOSu2019s copyright requirements here: PLOS Licenses and Copyright. If you need to request permissions from a copyright holder, you may use PLOS's Copyright Content Permission form.

Potential Copyright Issues:

i) Figure S1 File. Please confirm whether you drew the images / clip-art within the figure panels by hand. If you did not draw the images, please provide (a) a link to the source of the images or icons and their license / terms of use; or (b) written permission from the copyright holder to publish the images or icons under our CC BY 4.0 license. Alternatively, you may replace the images with open source alternatives. See these open source resources you may use to replace images / clip-art:

3) We note that your Data Availability Statement is currently as follows: "All the data used in this submission is contained within the manuscript.". Please confirm at this time whether or not your submission contains all raw data required to replicate the results of your study. Authors must share the “minimal data set” for their submission. PLOS defines the minimal data set to consist of the data required to replicate all study findings reported in the article, as well as related metadata and methods (https://journals.plos.org/plosone/s/data-availability#loc-minimal-data-set-definition).

4) Please ensure that the funders and grant numbers match between the Financial Disclosure field and the Funding Information tab in your submission form. Note that the funders must be provided in the same order in both places as well. Currently, the Financial Disclosure states there was no funding received.

**Reviewers' comments:**

Reviewer's Responses to Questions

**Key Review Criteria Required for Acceptance?**

**Methods**

-Are the objectives of the study clearly articulated with a clear testable hypothesis stated?

-Is the study design appropriate to address the stated objectives?

-Is the population clearly described and appropriate for the hypothesis being tested?

-Is the sample size sufficient to ensure adequate power to address the hypothesis being tested?

-Were correct statistical analysis used to support conclusions?

-Are there concerns about ethical or regulatory requirements being met?

Reviewer #1: (1) Could authors please comment on the comparative PCR: is this an in-house developed or commercial assay? Is it fully-validated or accredited? Why was this test used as comparison?

Reviewer #2: (No Response)

**Results**

-Does the analysis presented match the analysis plan?

-Are the results clearly and completely presented?

-Are the figures (Tables, Images) of sufficient quality for clarity?

Reviewer #1: The data are well presented.

Reviewer #2: (No Response)

**Conclusions**

-Are the conclusions supported by the data presented?

-Are the limitations of analysis clearly described?

-Do the authors discuss how these data can be helpful to advance our understanding of the topic under study?

-Is public health relevance addressed?

Reviewer #1: Minor:

(6) Line 40/41: “demonstrate the performance” in study C needs to be clarified further whether this is analytical performance.

(7) Line 54/55: “low resource laboratory settings”. This needs to be justified – e.g. costing, ease of use etc.

Reviewer #2: (No Response)

**Editorial and Data Presentation Modifications?**

Reviewer #1: (No Response)

Reviewer #2: (No Response)

**Summary and General Comments**

Reviewer #1: This is a very well-written study examining laboratory validation of a commercial cartridge-based PCR system on helminth detections. It has great relevance to the diagnostics accessibility in resource-limited settings.

I suggest the authors address the following points:

(1) To discuss at the introduction the context of this study in the absence of true “gold standard” (10.1016/j.ijpara.2014.05.009)

(2) Discuss limitations of this study: e.g. monitoring of load or intensity of disease, evaluation of treatment response, shedding dynamics are not in scope

(3) Line 69: “population-based surveys” have not been evaluated in this study, hence should not be included in the conclusion.

(4) Suggest discuss the PPV and NPV in the context of disease prevalence in Mozambique.

(5) Line 457-464: authors many want to consider rephrasing this as a general issue rather than a specific company issue, e.g. “a well-developed instrument support programme is required to ensure sustainable testing in Africa due to limited capability for instrument technical support by individual companies.”

Reviewer #2: (No Response)

PLOS authors have the option to publish the peer review history of their article (what does this mean? ). If published, this will include your full peer review and any attached files.

**Do you want your identity to be public for this peer review?** For information about this choice, including consent withdrawal, please see our Privacy Policy .

Reviewer #1: **Yes: ** Chuan Kok Lim

Reviewer #2: **Yes: ** Manfred Dakorah Asiedu

**Figure resubmission:**
---

## [Editor Report · Decision Letter 1]

31 Oct 2025

Dear DR. MESSA,

We are pleased to inform you that your manuscript 'Performance evaluation of the diaxxoPCR system for rapid and user-friendly stool-based diagnosis of trichuriasis, ascariasis and strongyloidiasis in Mozambique' has been provisionally accepted for publication in PLOS Neglected Tropical Diseases.

Best regards,

Angela Monica Ionica, Ph.D.

Academic Editor

Jong-Yil Chai

Section Editor

Shaden Kamhawi

co-Editor-in-Chief

Paul Brindley

co-Editor-in-Chief

---

## [Editor Report · Acceptance letter]

Dear Mr Messa Jr,

We are delighted to inform you that your manuscript, "Performance evaluation of the diaxxoPCR system for rapid and user-friendly stool-based diagnosis of trichuriasis, ascariasis and strongyloidiasis in Mozambique," has been formally accepted for publication in PLOS Neglected Tropical Diseases.

Best regards,

Shaden Kamhawi

co-Editor-in-Chief

Paul Brindley

co-Editor-in-Chief
